# REFRAMING DENSE ACTION DETECTION (REF-DENSE): A NEW PERSPECTIVE ON PROBLEM SOLVING AND A NOVEL OPTIMIZATION STRATEGY

## ABSTRACT

In dense action detection, we aim to detect multiple co-occurring actions. However, action classes are often ambiguous, as they share overlapping sub-components. We argue that the dual challenges of temporal and class overlaps are too complex to be effectively addressed as a single problem by a unified network. To overcome this, we propose decomposing the task into detecting temporally dense but unambiguous components underlying the action classes, and assigning these sub-problems to distinct sub-networks. By isolating unambiguous concepts, each sub-network focuses solely on resolving dense temporal overlaps, thereby simplifying the overall problem. Furthermore, co-occurring actions in a video often exhibit interrelationships, and exploiting these relationships can improve the method performance. However, current dense action detection networks fail to effectively learn these relationships due to their reliance on binary cross-entropy optimization, which treats each class independently. To address this limitation, we propose providing explicit supervision on co-occurring concepts during network optimization through a novel language-guided contrastive learning loss. Our extensive experiments demonstrate the superiority of our approach over state-of-the-art methods, achieving substantial improvements across different metrics on three challenging benchmark datasets, TSU, Charades, and MultiTHUMOS. Our code will be released upon paper publication.

## 1 INTRODUCTION

Dense action detection aims to recognize and temporally localize all actions within an untrimmed video, even when multiple actions occur concurrently. A deep understanding of these complex semantics is crucial for real-world applications, such as autonomous driving, sports analytics, and complex surveillance, where actions rarely occur in isolation. To address this task, current approaches (*e.g.*, Dai et al. (2019); Tirupattur et al. (2021); Dai et al. (2022a; 2023); Sardari et al. (2023); Zhu et al. (2024)) typically follow a common pipeline. First, features of the video's frames are extracted using a pre-trained encoder. These features are then passed through a temporal modeling block to capture dependencies across time, followed by a classification head that maps the learned temporal representations to multi-label action probabilities over time—enabling dense action detection. The entire network is optimized using Binary Cross-Entropy (BCE) loss.

In dense action detection, beyond the challenge of temporal overlaps, action classes often exhibit overlapping components (*i.e.*, class ambiguity). This overlap can arise from shared entities or motions that define the action classes. For example, in the MultiTHUMOS dataset (Yeung et al. (2018)), the action classes "Hammer Throw Wind Up" and "Hammer Throw Spin" share an identical entity, a hammer. Similarly, in the Charades dataset (Sigurdsson et al. (2016)), the classes "Holding a Bag" and "Holding a Sandwich" overlap in motion, the act of holding. We argue that addressing both temporal and class overlaps simultaneously as **a single problem**—as done in traditional pipelines—is inherently too complex for a unified network. This motivated our core question: **Can we reduce the problem's complexity by eliminating class overlaps, allowing the network to focus solely on resolving temporal overlaps?** To this end, we introduce a new perspective on solving this task. Rather than directly detecting dense, ambiguous actions with a single unified network, we propose decomposing the task into detecting temporally dense but unambiguous sub-components underly-

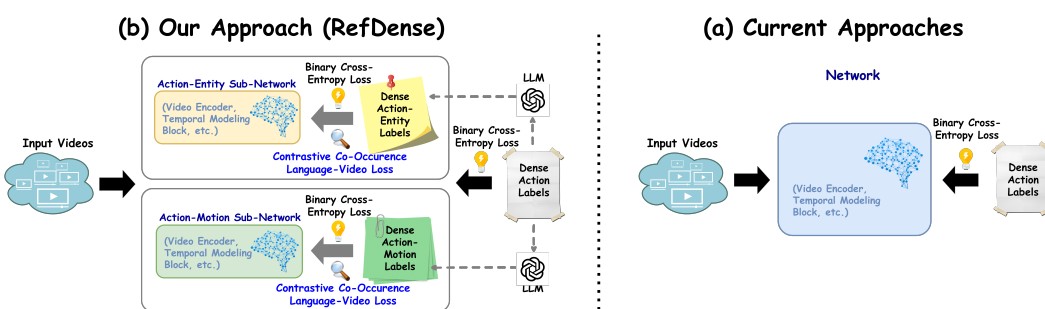

Figure 1: Comparison of current approaches and our proposed approach, RefDense, for tackling the dense action detection task. (a) Current approaches address the entire task as a single problem (*i.e.*, detecting dense, ambiguous actions) using a unified network optimized solely with Binary Cross-Entropy (BCE) loss. In contrast, (b) RefDense decomposes the task into two sub-problems (*i.e.*, detecting dense, unambiguous entity and motion sub-components underlying the actions classes) and assigns them to distinct sub-networks. Furthermore, our approach is optimized using both BCE loss and our proposed contrastive co-occurrence language-video loss.

ing the action classes (*i.e.*, entity and motion components), and assigning these sub-problems to distinct sub-networks. By isolating the unambiguous components of actions, each sub-network can concentrate exclusively on resolving one single challenge—dense temporal overlaps.

To implement our proposed perspective for solving dense action detection, we (i) design a framework comprising two sub-networks, Action-Entity and Action-Motion, and (ii) decompose dense temporal action labels into dense temporal action-entity and dense temporal action-motion labels using prompts and a pre-trained large language model (LLM). While both sub-networks receive the same input video—and can potentially share the same architecture—Action-Entity is dedicated to detecting dense temporal entity components involved in actions, whereas Action-Motion focuses on detecting dense temporal motion components. The dense temporal entity and motion representations learned by the sub-networks are then concatenated for final dense action detection. For brevity, we omit the term temporal and refer to "dense temporal" labels and concepts simply as "dense" for the remainder of the paper.

In dense action detection, where multiple concepts can occur simultaneously, awareness of class dependencies can significantly enhance performance. For instance, in scenarios like cooking, actions such as "Pouring" and "Stirring" often occur together. However, we argue that the current dense action detection networks (*e.g.*, Tirupattur et al. (2021); Sardari et al. (2023); Zhu et al. (2024)) cannot effectively learn the relationships among the co-occurrence classes as they are trained using the BCE loss which treats each action class independently during the optimization process. This limitation motivates us to raise our second novel question: **Can we improve network optimization to fully unlock the potential benefits of co-occurring concepts?** To achieve this, we propose providing explicit supervision on co-occurring concepts in the input video during network optimization. Inspired by contrastive language-image pretraining (Radford et al. (2021)), we introduce Contrastive Co-occurrence Language-Video learning, which aligns the video features in the embedding space with the textual features of all co-occurring classes. Specifically, we assign a textual description to each co-occurring concept in the input video and use a frozen, pre-trained text encoder to extract its features. We then adapt the noise contrastive estimation (NCE) loss to match the video features with the text features of all co-occurring classes. Through this approach, the network not only receives explicit guidance about co-occurring concepts during training, but also implicitly benefits from the learned semantic among related concepts within the embedding space of pre-trained language models. In Fig. 1, we compare current approaches to our proposed method (RefDense) in tackling the dense action detection task.

Our key contributions are summarized as follows: (i) We identify the challenge of simultaneous temporal and class overlaps in dense action detection—an aspect that has not been explicitly explored in prior work—which opens new opportunities for future research. (ii) To address this challenge, we propose a novel problem-solving perspective, *i.e.*, decomposing the problem complexity for the network. This approach can also benefit other dense computer vision tasks, *e.g.*, dense captioning. (iii)

We pioneer an optimization strategy that explicitly leverages supervision on co-occurring concepts during training. This can improve the performance of existing and future dense action detection networks. (iv) Our comprehensive comparison using multiple metrics and three challenging benchmark datasets against state-of-the-art approaches demonstrates the superiority of our method. (v) Our extensive ablation studies evaluated across multiple metrics, highlight the effectiveness of each component in our method's design.

## 2 RELATED WORKS

**Dense Action Detection –** Current dense action detection approaches (*e.g.*, Dai et al. (2019); Tirupattur et al. (2021); Sardari et al. (2023); Dai et al. (2023); Zhu et al. (2024)) typically follow a common pipeline. First, the video is divided into segments, and a frozen, pre-trained encoder (*e.g.*, I3D Carreira & Zisserman (2017), CLIP Radford et al. (2021)) extracts features from each segment. These features are then passed to a temporal modeling block that captures their temporal relationships, followed by a classification layer that maps the learned representations to multi-action probabilities. The entire network is optimized using the BCE loss. Although most of the pipeline is shared across approaches, the primary distinctions lie in the design of the temporal modeling block. Below, we briefly review this block in existing approaches.

Pre-transformer approaches, such as Piergiovanni & Ryoo (2018; 2019); Kahatapitiya & Ryoo (2021), rely on Gaussian or convolutional filters to represent a video as a sequence of multi-activity events. While these methods are effective at modeling short, dense actions, the inherent temporal limitations of Gaussian and convolutional kernels restrict their ability to capture longer actions. With the success of transformers in modeling long-term dependencies, several works (Tirupattur et al. (2021); Dai et al. (2022a); Sardari et al. (2023); Dai et al. (2021b; 2023); Zhu et al. (2024)) have developed transformer-based networks. Among these, some approaches, such as Dai et al. (2022a); Sardari et al. (2023); Tan et al. (2022); Zhu et al. (2024), focus on modeling various ranges of temporal relationships using multi-scale transformer networks or DETR-based architectures (Carion et al. (2020)). On the other hand, Tirupattur et al. (2021) introduce the concept of benefiting from learning co-occurrence class relationships. To learn these relationships, they propose explicitly modeling all action classes within the network architecture. Similarly, Dai et al. (2023) embed all objects in the dataset into the network's architecture. However, not only do their designs lack computational efficiency due to their dependence on the maximum number of classes, but they also fail to fully capture co-occurrence relationships despite explicitly modeling the classes, as the networks are still optimized using the BCE loss, which treats each class independently. To the best of our knowledge, for the first time, our proposed contrastive co-occurrence language-video loss, is designed to overcome this limitation in network optimization by providing explicit supervision on co-occurring concepts during training. Furthermore, as it is a general loss function applied in the embedding space, it can benefit the optimization process in any existing or future network.

Although transformer-based approaches show performance improvements over traditional methods, the inherent complexity of handling both temporal and action class overlaps poses a substantial obstacle for networks. We addresses this by eliminating one of the overlaps; we propose to decompose the task of detecting dense ambiguous actions to detecting dense non-ambiguous components underlying the action classes, and assign these sub-problems to distinct sub-networks. By isolating these non-ambiguous components, each sub-network focuses exclusively on resolving a single challenge, dense temporal overlaps.

**Two-Stream Networks –** Two-stream approaches (Simonyan & Zisserman (2014); Carreira & Zisserman (2017)) model spatial and temporal information using separate modalities or architectures and are widely used in video understanding. Despite this separation, they often fail to effectively capture high-level entity and motion semantics, as they are optimized end-to-end on the overall action-detection task without explicit semantic supervision. In sparse-label scenarios—where at most one action occurs per timestamp—streams may gradually acquire these semantics implicitly. However, in dense multi-label settings, where multiple actions overlap, end-to-end optimization alone struggles to disentangle them. In contrast, our approach explicitly learns high-level semantic concepts within each stream using dense temporal action-entity and action-motion sub-label supervision derived from the original annotations.

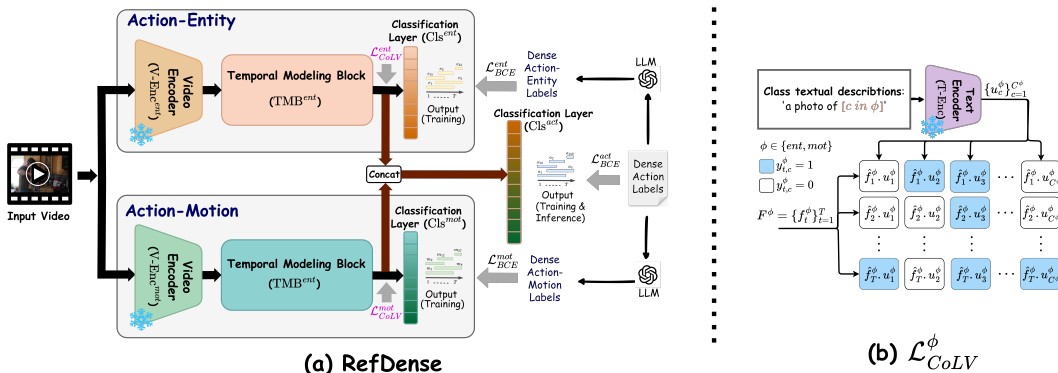

(a) RefDense

(b) $\mathcal{L}_{CoLV}^{\phi}$

Figure 2: (a) The overall scheme of RefDense. Our proposed network consists of two sub-networks: Action-Entity and Action-Motion. Action-Entity learns dense entity components associated with the action classes, while Action-Motion focuses on learning dense motion components related to the action classes. The entire network is optimized using the dense action labels and the BCE loss ($\mathcal{L}_{BCE}^{act}$). Additionally, the sub-networks are optimized using dense action-entity and action-motion labels, which are derived from action labels, along with the BCE loss ($\mathcal{L}_{BCE}^{ent}$, and $\mathcal{L}_{BCE}^{mot}$) and our proposed contrastive co-occurrence language-video loss ($\mathcal{L}_{CoLV}^{ent}$ and $\mathcal{L}_{CoLV}^{mot}$). (b) Alignment of temporal video features with textual features of co-occurring class concepts in $\mathcal{L}_{CoLV}^{ent}$ and $\mathcal{L}_{CoLV}^{mot}$.

**Vision-Language for Action Detection –** Building on CLIP's zero-shot capabilities (Radford et al. (2021)), many works, such as Nag et al. (2022); Li et al. (2024); Fish et al. (2024); Liberatori et al. (2024), adapt its language-image pre-training paradigm for zero-shot or few-shot action detection. Following this, some works, such as Cao et al. (2022); Xu et al. (2022), explore using language models for network pre-training. In contrast, Ju et al. (2023); Dai et al. (2023) integrate language models directly during training. For instance, Dai et al. (2023) introduce an object-centric graph for indoor activity detection and leverage language supervision to ensure that each graph node corresponds to a distinct object, while Ju et al. (2023) use language to obtain pseudo-labels for weakly supervised learning. In a similar spirit, we benefit from language models during training. However, our goal is different from that of prior works; we aim to leverage language to effectively learn the relationships among co-occurring concepts.

## 3 METHODOLOGY

In this section, we first define the dense action detection task and briefly review the common pipeline used by current approaches to tackle this task. We then elaborate on our proposed approach, Ref-Dense.

### 3.1 PRELIMINARIES

**Problem Definition –** In the dense action detection task, the goal is to identify all actions occurring at each timestamp of an untrimmed video, as described in (Tirupattur et al. (2021); Dai et al. (2022a); Sardari et al. (2023); Zhu et al. (2024)). Given an untrimmed video sequence $V = \{I_n \in \mathbb{R}^{W \times H \times 3}\}_{n=1}^{N}$ of length $N$, each timestamp $n$ has a multi-action class label $Y_n = \{y_{n,c} \in \{0,1\}\}_{c=1}^{C^{act}}$, where $C^{act}$ represents the total number of action classes in the dataset, and the set of action labels for the entire video is denoted as $Y = \{Y_n\}_{n=1}^{N}$. The network's task is to estimate multi-action class probabilities $P = \{P_n\}_{n=1}^{N}$, where $P_n = \{p_{n,c} \in [0,1]\}_{c=1}^{C^{act}}$.

**Current Pipeline to Tackle Dense Action Detection –** The most widely used pipeline for dense action detection consists of three main components: a Video Encoder, a Temporal Modeling Block, and a Classification layer. First, the Video Encoder—typically a frozen pre-trained I3D (Carreira & Zisserman (2017)) or CLIP image encoder (Radford et al. (2021))—processes the input video sequence $V$ for the Temporal Modeling Block. The video is divided into non-overlapping $K$-frame video segments $V = \{S_t\}_{t=1}^{T}$, where $S_t \in \mathbb{R}^{K \times W \times H \times 3}$ and $T = \frac{N}{K}$. These segments are then fed

into the encoder to obtain segment-level input tokens:

$$\hat{F} = \{\text{V-Enc}(S_t)\}_{t=1}^T, \text{ where } \hat{F} \in \mathbb{R}^{\text{T} \times \text{D}}. \tag{1}$$

Next, the Temporal Modeling Block receives the segment-level tokens and captures temporal relationships among them:

$$F = \text{TMB}(\hat{F}), \text{ where } F \in \mathbb{R}^{\text{T} \times \text{D}^*}. \tag{2}$$

Different approaches implement this block using different architectures, such as Graph Neural Networks (GNNs) (Dai et al. (2021a; 2023)) or multi-scale transformers (Dai et al. (2022a); Sardari et al. (2023)). Finally, the Classification layer—typically composed of fully connected layers or 1D convolutional filters—is applied to the output of the Temporal Modeling Block to produce multi-action class probabilities for all temporal segment $P = Cls(F)$, where $P \in \mathbb{R}^{\text{T} \times \text{C}^{\text{act}}}$. The entire network is typically optimized using the ground truth labels $Y$ and the BCE loss as

$$\mathcal{L}_{BCE} = BCE(Y, P) = -\frac{1}{T} \sum_{t=1}^{T} \sum_{c=1}^{C^{act}} \ell_{\text{bce}}(y_{t,c}, p_{t,c}), \tag{3}$$

$$\ell_{\text{bce}}(y, p) = y \log(p) + (1 - y) \log(1 - p). \tag{4}$$

### 3.2 Reframing Dense Action Detection (RefDense)

We introduce a new perspective on solving the dense action detection task. Instead of tackling the entire complex problem—handling the dual challenge of temporal and action class overlaps (*i.e.*, class ambiguity)—with a single unified network, we propose decomposing the problem into less complex sub-problems: detecting temporally dense, but unambiguous components underlying action classes (*i.e.*, entity and motion components) and assigning these sub-problems to distinct sub-networks. By isolating these unambiguous components of actions, the sub-networks can focus exclusively on resolving a single challenge—dense temporal overlaps.

To implement our proposed perspective, we (i) design a framework comprising two sub-networks: Action-Entity and Action-Motion, and (ii) decompose dense action labels into dense action-entity and dense action-motion labels using prompts and a pre-trained LLM. Our proposed approach is illustrated in Fig. 2(a). While both sub-networks receive the same input video, Action-Entity focuses on detecting dense entity concepts, whereas Action-Motion is dedicated to detecting dense motion concepts involved in the input video's actions. The dense temporal entity and motion representations learned by the sub-networks are then concatenated for dense action detection. The entire network is optimized using dense action labels and the BCE loss, while the Action-Entity and Action-Motion sub-networks are also individually optimized using dense action-entity and dense action-motion labels, respectively, with the BCE loss. Furthermore, to effectively leverage the interrelationships among co-occurring concepts within the video, we optimize the network's embedding space using our proposed contrastive co-occurrence language-video loss. In the following, we detail our framework, label decomposition, and loss functions.

**Dense Action-Entity & Dense Action-Motion Labels –** The sub-labels are derived for each input video from its original ground-truth action labels. To this end, first, a set of action-entity classes $\mathcal{E} = \{e_c\}_{c=1}^{C^{ent}}$ and action-motion classes $\mathcal{M} = \{m_c\}_{c=1}^{C^{mot}}$ are defined from the full set of action classes $\mathcal{A} = \{a_c\}_{c=1}^{C^{act}}$ in the dataset using a specific prompt and a pre-trained large language model, GPT-4 (see Appendix for details). For instance, from the action class "Weight Lifting Clean", the action-entity class "Barbell" and the action-motion class "Lifting-Clean" are extracted, respectively. Since some action classes have overlapping entity and motion components, the number of derived classes are less than the action classes, $C^{ent}, C^{mot} < C^{act}$. Then, for each input video, using its corresponding action ground-truth label $Y$ and the newly defined classes, we generate its dense action-entity and dense action-motion labels $Y^{ent} = \{Y_t^{ent}\}_{t=1}^T$ and $Y^{mot} = \{Y_t^{mot}\}_{t=1}^T$ as:

$$Y_t^{ent} = \{y_{t,c}^{ent}\}_{c=1}^{C^{ent}}, \ y_{t,c}^{ent} = \max_{j=1,\dots,C^{act}} \left( y_{t,j} \cdot \mathbf{1}_{\{e_c \in \text{entities}(a_j)\}} \right), \tag{5}$$

$$Y_t^{mot} = \{y_{t,c}^{mot}\}_{c=1}^{C^{mot}}, \ y_{t,c}^{mot} = \max_{j=1,\dots,C^{act}} \left( y_{t,j} \cdot \mathbf{1}_{\{m_c \in \text{motions}(a_j)\}} \right), \tag{6}$$

where $\mathbf{1}_{\{.\}}$ is the indicator function that checks if action-entity class $e_c$ or action-motion class $m_c$ is associated with action class $a_j$. It is equal to 1 if the condition is true and 0 otherwise. We would

like to note that not all actions involve both entities and motion components (e.g., "Walking"). For actions with only one component, the sub-label is assigned solely to that component.

**Action-Entity Sub-Network –** This sub-network is designed to detect dense entity concepts present in the action classes. First, dense temporal action–entity representations are extracted by passing the video segments through a frozen pre-trained encoder followed by a temporal modeling block, $F^{ent} = TMB^{ent}(\{\text{V-Enc}^{ent}(S_t)\}_{t=1}^T)$. Next, a classification layer is applied to these representations to predict multi-entity probabilities for all video segments, $P^{ent} = \text{Cls}^{ent}(F^{ent})$, where $P^{ent} \in \mathbb{R}^{T \times C^{ent}}$.

**Action-Motion Sub-Network –** This sub-network instead focuses on capture dense motion concepts associated with the action classes. To do so, each video segment is first processed through a frozen pre-trained Encoder, followed by a Temporal Modeling Block to learn dense temporal action–motion representations, $F^{mot} = TMB^{mot}(\{\text{V-Enc}^{mot}(S_t)\}_{t=1}^T)$. These representations are then passed through a classification layer to estimate multi-motion probabilities for all video segments, $P^{mot} = \text{Cls}^{mot}(F^{mot})$, where $P^{mot} \in \mathbb{R}^{T \times C^{mot}}$.

**Sub-Networks Fusion for Dense Action Detection –** To perform dense action detection, the dense entity and motion video representations learned by the sub-networks are first concatenated to form the full video representation $F^{act} = [F^{ent}; F^{mot}]$, where $[;]$ denotes the concatenation operation. Then, a Classifier layer is applied to the full features to predict multi-action probabilities for all video segments as $P^{act} = \text{Cls}^{act}(F^{act})$, where $P^{act} \in \mathbb{R}^{T \times C^{act}}$.

**Binary Cross-Entropy Optimization –** With the action probabilities $P$ and action ground-truth labels $Y$, the entire network is optimized using $\mathcal{L}_{BCE}^{act} = BCE(Y, P^{act})$. The Action-Entity and Action-Motion sub-networks are also individually optimized using BCE and dense action-entity and action-motion ground-truth labels $Y^{ent}$ and $Y^{mot}$, respectively, as $\mathcal{L}_{BCE}^{ent} = BCE(Y^{ent}, P^{ent})$ and $\mathcal{L}_{BCE}^{mot} = BCE(Y^{mot}, P^{mot})$.

**Contrastive Co-Occurrence Language-Video Learning –** In scenarios where multiple concepts occur simultaneously, awareness of class dependencies can improve performance. However, we argue that optimizing with BCE loss does not effectively capture these relationships, as BCE treats each class label independently. To address this, we propose providing explicit supervision on co-occurring concepts during training. Inspired by contrastive language–image pre-training (Radford et al. (2021)), we align the learned video representations $F^\phi = \{f_t^\phi\}_{t=1}^T$ in the embedding space with the text features of all co-occurring classes in the input video (see Fig. 2(b)). Specifically, for each class $c$ in the class set $\phi$, we construct a textual description $txt_c^\phi = $ 'a photo of $[c \text{ in } \phi]$', where $[c \text{ in } \phi]$ denotes the natural-language description of class $c$ within the class set $\phi$. Next, a frozen pre-trained Text Encoder extracts their features $u_c^\phi = \text{T-Enc}(txt_c^\phi)$. Finally, the noise contrastive estimation is adapted to match the visual representation in the $t^{th}$ video segment, with the text features of all the co-occurring concepts in that segment as:

$$\mathcal{L}_{CoLV} = \sum_\phi \mathcal{L}_{CoLV}^\phi, \tag{7}$$

$$\mathcal{L}_{CoLV}^\phi = -\frac{1}{T} * \sum_{t=1}^T \frac{1}{|\beta(t)^\phi|} \sum_{b \in \beta(t)^\phi} \log \frac{\exp(\hat{f}_t^{\phi\top}.u_b^\phi/\tau)}{\sum_{\substack{c=1, \\ c \notin \beta(t)^\phi}}^{C^\phi} \exp(\hat{f}_t^{\phi\top}.u_c^\phi/\tau)}, \tag{8}$$

$$\beta(t)^\phi = \{b \mid b \in \{1, 2, ..., C^\phi\}, y_{t,b}^\phi = 1\}, \ \phi \in \{ent, mot\}. \tag{9}$$

Through this, the network not only receives explicit knowledge of co-occurring concepts, but also implicitly benefits from the learned semantic among related concepts within the embedding space of pre-trained language models.

## 4 EXPERIMENTAL RESULTS

**Datasets –** We evaluate our proposed approach on all three benchmark datasets for this task: TSU (Dai et al. (2022b)), Charades (Sigurdsson et al. (2016)), and MultiTHUMOS (Yeung et al. (2018)). TSU and Charades contain 536 and 9,848 videos of daily activities, respectively, covering 51 and

Table 1: Dense action detection results on the TSU, Charades and MultiTHUMOS datasets using RGB inputs, in terms of per-frame mAP. The best and the second best results are in **Bold** and underlined. † indicates results produced by running the authors' publicly available code.

| Method | | GFLOPs | V-Enc | mAP(%) | | |
|---|---|---|---|---|---|---|
| | | | | TSU | Charades | MultiTHUMOS |
| SuperEvent (Piergiovanni & Ryoo, 2018) | CVPR | 0.8 | I3D | 17.2 | 18.6 | 36.4 |
| TGM (Piergiovanni & Ryoo, 2019) | ICML | 1.2 | I3D | 26.7 | 20.6 | 37.2 |
| PDAN (Dai et al., 2021b) | WACV | 3.2 | I3D | 32.7 | 23.7 | 40.2 |
| CoarseFine (Kahatapitiya & Ryoo, 2021) | CVPR | - | X3D | - | 25.1 | - |
| MLAD (Tirupattur et al., 2021) | CVPR | 44.8 | I3D | - | 18.4 | 42.2 |
| CTRN (Dai et al., 2021a) | BMVC | - | I3D | 33.5 | 25.3 | 44.0 |
| PointTAD (Tan et al., 2022) | NeurIPS | - | I3D | - | 21.0 | 39.8 |
| MS-TCT (Dai et al., 2022a) | CVPR | 6.6 | I3D | 33.7 | 25.4 | 43.1 |
| HAAN (Gao et al., 2023) | ICM | - | I3D | - | 25.1 | 41.7 |
| PAT (Sardari et al., 2023) | ICCVW | 8.5 | I3D | 34.0† | 26.5 | 44.6 |
| DualDET (Zhu et al., 2024) | CVPR | 5.5 | I3D | 34.8 | 23.2 | 45.5 |
| **RefDense** | | 11.4 | I3D | **36.4** (+1.4) | **26.9** (+0.4) | **46.8** (+1.3) |
| TTM (Ryoo et al., 2023) | CVPR | - | ViViT | - | 28.8 | - |
| MS-TCT (Dai et al., 2022a)† | CVPR | 6.6 | CLIP | 39.2 | 32.1 | 43.3 |
| PAT (Sardari et al., 2023)† | ICCVW | 8.5 | CLIP | 40.8 | 33.1 | 44 |
| ANN (Dai et al., 2023) | BMVC | - | CLIP | 41.3 | 32.0 | - |
| **RefDense** | | 11.4 | CLIP | **43.4** (+2.1) | **34.1** (+1.0) | 45.4 (+1.4) |

157 action classes with a high degree of temporal overlap among action instances. The videos in TSU are long with an average length of 21 minutes, while Charades consists of short videos with an average length of 30 seconds. MultiTHUMOS, the dense multi-label extension of the single-label action detection dataset THUMOS'14 (Jiang et al. (2014)), includes 413 sports videos spanning 65 action classes, with an average length of 3.5 minutes as reported in (Zhu et al. (2024)).

**Implementation Details –** To implement the Action-Entity and Action-Motion sub-networks, we adopt PAT (Sardari et al. (2023)), a network that employs a non-hierarchical multi-scale transformer for dense detection. For Action-Motion, we retain the original 4-level temporal scale blocks, while for Action-Entity, only the first-level block is used, as entity detection is comparatively simpler and does not warrant additional computational overhead. The length of each video segment is set to $K = 8$ frames. During training, $T$ is set to 2500, 256, and 800 for TSU, Charades, and MultiTHUMOS, respectively, while at inference, we follow previous work (Tirupattur et al. (2021); Kahatapitiya & Ryoo (2021); Sardari et al. (2023)) and make predictions on the full video sequence. For TSU, 31 action-entity and 28 action-motion classes are defined; for Charades, 38 action-entity and 38 action-motion classes; and for MultiTHUMOS, 28 action-entity and 50 action-motion classes are defined. The coefficients for $(\mathcal{L}_{CoLV}^{ent}, \mathcal{L}_{CoLV}^{mot})$ and $(\mathcal{L}_{BCE}^{ent}, \mathcal{L}_{BCE}^{mot})$ and $\mathcal{L}_{BCE}^{act}$ are set to 0.3, 0.1, and 1.0, respectively, which were determined through trial and error. See more details in Appendix.

### 4.1 STATE-OF-THE-ART COMPARISON

In this section, we compare the performance of our approach with current state-of-the-art methods. Note: Here, our results and comparisons are based on RGB input features. However, results and comparisons incorporating RGB and optical flows can be found in the Appendix.

The primary metric for dense action detection task is per-frame mAP. Table 1 presents comparative results using this metric. The results demonstrate the superiority of our approach over state-of-the-art methods, achieving over 1.0% improvement on most benchmarks, including 1.4% and 2.1% gains on the TSU dataset, where action instances have significant temporal and semantic overlap.

The standard mAP assesses the performance by evaluating each class independently. However, it does not explicitly measure whether models learn the relationships amongst the classes. To overcome this, Tirupattur et al. (2021) introduce a set of action-conditional metrics, including action-conditional mean Average Precision ($mAP_{ac}$), action-conditional F1-Score ($F1_{ac}$), action-conditional Precision ($P_{ac}$), and action-conditional Recall ($R_{ac}$). These metrics aim to explicitly assess how well pairwise class/action dependencies are modeled, both within a single frame and across different frames. Table 2 presents the comparative results on TSU and Charades using action-conditional metrics. While these metrics evaluate a method's performance more effectively than

Table 2: Dense action detection results on Charades and MultiTHUMOS using RGB inputs and CLIP as V-Enc's backbone, evaluated based on the action-conditional metrics with cross-action dependencies over a temporal window of size $\delta$. The best and the second best results are in **Bold** and underlined. † indicates results produced by running the authors' publicly available code.

| Method | TSU | | | | | | | | Charades | | | | | | | |
| | $\delta = 0$ | | | | $\delta = 20$ | | | | $\delta = 0$ | | | | $\delta = 20$ | | | |
| | mAP$_{ac}$ | F1$_{ac}$ | P$_{ac}$ | R$_{ac}$ | mAP$_{ac}$ | F1$_{ac}$ | P$_{ac}$ | R$_{ac}$ | mAP$_{ac}$ | F1$_{ac}$ | P$_{ac}$ | R$_{ac}$ | mAP$_{ac}$ | F1$_{ac}$ | P$_{ac}$ | R$_{ac}$ |
|---|---|---|---|---|---|---|---|---|---|---|---|---|---|---|---|---|
| MS-TCT† | 29.5 | 27.0 | 20.6 | 38.0 | 42.2 | 38.3 | 31.7 | 46.5 | 36.7 | 17.1 | 28.1 | 12.2 | 42.4 | 18.9 | 28.4 | 14.15 |
| PAT† | 30.6 | 27.8 | 21.9 | 38.2 | 44.0 | 40.1 | 35.1 | 46.7 | 37.7 | 31.7 | 31.0 | 32.7 | 44.0 | 35.9 | 35.1 | 37.0 |
| ANN | - | - | - | - | - | - | - | - | 35.4 | 20.4 | 31.4 | - | 41.8 | 22.3 | 30.4 | - |
| **RefDense** | **33.2** | **30.3** | **24.9** | **38.6** | **46.7** | **42.7** | **39.0** | **47.0** | **38.6** | **33.1** | **32.1** | **34.2** | **44.5** | **36.9** | **35.6** | **38.3** |
| | (+2.6) | (+2.5) | (+3.0) | (+0.4) | (+2.7) | (+2.6) | (+4.1) | (+0.3) | (+0.9) | (+2.4) | (+0.7) | (+1.5) | (+0.5) | (+1.0) | (+0.6) | (+1.3) |

Figure 3: Qualitative comparison with previous approaches (PAT (Sardari et al. (2023)) and MS-TCT (Dai et al. (2022a))) on a test video sample of TSU.

standard mAP, only a few methods report results using them, primarily on Charades. Therefore, for a comprehensive comparison, we produced the results of previous methods under these metrics, using RGB inputs, with their publicly available code whenever accessible. Table 2 demonstrates the superiority of our method over current state-of-the-art approaches in detecting dense actions. Specifically, it achieves an average improvement of **2.3%** on TSU and **1.2%** on Charades across all metrics.

**Qualitative Comparison –** In Fig. 3, we qualitatively compare our approach with the state-of-the-art methods PAT (Sardari et al. (2023)) and (MS-TCT Dai et al. (2022a)). The results show that, on average, our method's predictions align more closely with the ground-truth labels than those of other methods. In particular, it detects more action types than MS-TCT (*i.e.*, it fails to detect the action "Drinking from a cup"). Compared with PAT, which correctly identifies the true action types present in the video as our method does, PAT struggles with class ambiguities. For instance, for the actions "Drinking from a cup" and "Drinking from a can", which share overlapping motion components, "Drinking", PAT misclassifies the latter and produces a false positive prediction, whereas our method correctly distinguishes between them.

## 4.2 ABLATION STUDIES

In this section, we evaluate the impact of key components of our proposed approach using both type of metrics and on the TSU dataset. Note, to perform these experiments, CLIP is used as V-Enc backbone, and all action conditional metrics are measured over a temporal window of size $\delta = 0$.

**Impact of Task Decomposition for Network –** To evaluate the effect of our decomposition strategy, we compare the performance of RefDense in two settings: (i) the entire framework is trained only on the task of dense action detection, directly addressing the dual challenge of temporal and class overlaps without decomposition, and (ii) the sub-networks are additionally tasked with solving dense action-entity and dense action-motion detection. Our results in Table 3 demonstrate that our proposed approach—decomposing dense action detection into detecting unambiguous sub-components—achieves significant improvements of over **2%** on the mAP metric and an average gain of **1.6%** across all metrics.

**Impact of Our Proposed Optimization –** To assess the effect of our proposed optimization, we ablate our proposed contrastive co-occurrence language-video loss $\mathcal{L}_{CoLV}$ in Table 4. The results

Table 3: Impact of task decomposition for the network

| Network | Task(s) Solved | mAP | mAP$_{ac}$ | F1$_{ac}$ |
|---|---|---|---|---|
| RefDense | Dense Action Detection | 42.3 | 31.8 | 29.0 |
| RefDense | Dense Action Detection + (Dense Action-Entity Detection & Dense Action-Motion Detection) | **43.4** (+1.1) | **33.2** (+1.4) | **30.3** (+1.3) |

Table 4: Ablation studies on $\mathcal{L}_{CoLV}$.

| $\mathcal{L}_{CoLV}$ | mAP | mAP$_{ac}$ | F1$_{ac}$ |
|---|---|---|---|
| X | 42.4 | 31.5 | 29.5 |
| ✓ | **43.4** (+2.0) | **33.2** (+1.7) | **30.3** (+0.7) |

Table 5: Generalization of our proposed approach. ⊛ indicates that the network is embedded in our framework.

| Network | # Param (M) | mAP | mAP$_{ac}$ | F1$_{ac}$ |
|---|---|---|---|---|
| PAT | 270 | 38.7 | 29.5 | 27.3 |
| PAT⊛ | 144 | **42.8** (+4.1) | **32.4** (+2.9) | **29.1** (+1.8) |
| MS-TCT | 328 | 40.1 | 30.1 | 24.3 |
| MS-TCT⊛ | 140 | **41.4** (+1.3) | **31.0** (+0.9) | **28.8** (+4.5) |

Table 6: Generalization of our proposed loss, $\mathcal{L}_{CoLV}$.

| | mAP | mAP$_{ac}$ | F1$_{ac}$ |
|---|---|---|---|
| PAT | 41.1 | 31.2 | 29.0 |
| PAT + $\mathcal{L}_{CoLV}$ | **42.3** (+1.2) | **33.0** (+1.8) | **30.0** (+1.0) |
| MS-TCT | 39.2 | 29.5 | 27.0 |
| MS-TCT + $\mathcal{L}_{CoLV}$ | **40.2** (+2.0) | **31.4** (+1.9) | **29.0** (+2.0) |

indicate that providing explicit supervision on co-occurring concepts through our loss significantly enhances the method's performance, on average **1.5%** improvement across all metrics. Notably, this improvement is achieved purely through optimization, without modifying the network.

**Generalization of Our Proposed Approach –** Our approach introduces a general problem formulation for dense action detection, enabling any existing or future model to be applied within our perspective to address the dual challenges of temporal and class overlaps. In Table 5, we present two examples—PAT (Sardari et al. (2023)) and MS-TCT (Dai et al. (2022a))—and compare their performance under two settings: (i) training each model as a single unified network for the dense action detection task, and (ii) embedding them as Action-Entity and Action-Motion sub-networks within our proposed framework. To ensure fairness, both the unified and dual-branch frameworks use the same total embedding dimensionality; each branch in our framework has half the embedding size of the unified network. The results demonstrate that our proposed approach significantly enhances the performance of both PAT and MS-TCT, with an average improvement of more than **2.0%** across all metrics.

**Generalization of Our proposed Optimization –** Our proposed loss, $\mathcal{L}_{CoLV}$, is a general loss function that can be applied to the embedding space of any existing or future network, providing explicit supervision on co-occurring concepts during training. For example, in Table 6, we show its impact when applied to two existing networks, PAT (Sardari et al. (2023)) and MS-TCT (Dai et al. (2022a)). The results demonstrate that it significantly enhances their performance, achieving an average gain of 1.3% and 1.9% across all metrics for PAT and MS-TCT, respectively. Importantly, this improvement is achieved purely through optimization, without modifying the network architecture.

## 5 CONCLUSION

In this paper, we introduce a new perspective in solving the dense action detection task. Instead of tackling the entire complex problem—handling the dual challenge of temporal and action class overlaps (*i.e.*, class ambiguity)— as a single problem using a unified network, we propose decomposing the problem of detecting dense, ambiguous actions into detecting dense, unambiguous sub-components that define the action classes, and assigning these sub-problems to distinct sub-networks. By isolating these unambiguous concepts, each sub-network can focus exclusively on resolving a single challenge—dense temporal overlaps. Furthermore, to effectively learn the relationships among co-occurring concepts in a video, we propose a novel contrastive language-guided loss that provides explicit supervision on co-occurring concepts during training. Our extensive experiments, conducted on several challenging benchmark datasets using multiple metrics, demonstrate that our method significantly outperforms state-of-the-art approaches across all metrics. Additionally, ablation studies highlight the effectiveness of the key components of our method. Future work will extend our approach to dense multi-modal (*e.g.*, audio-visual) dense action detection.

## ETHICS STATEMENT

This work does not involve human subjects or crowdsourcing, and it does not use or curate data that contain personally identifiable information or offensive content. We confirm that we have read and complied with the ethics review guidelines for ICLR submissions.

## REPRODUCIBILITY STATEMENT

We use publicly available datasets: TSU (Dai et al. (2022b)), Charades (Sigurdsson et al. (2016)), and MultiTHUMOS (Yeung et al. (2018)). We will publicly release our dense action–entity and dense action–motion annotations for all three datasets, as well as our code, upon publication of the paper.

## LARGE LANGUAGE MODELS STATEMENT

We only used Large Language Models (LLMs) for grammar checking and polishing the writing.

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

# A APPENDIX

## A.1 EXTRACTING ACTION-ENTITY AND ACTION-MOTION LABELS

Action-entity and action-motion labels were extracted using our designed prompt and GPT-4, as illustrated in Fig. 4. While the label decomposition is automated using GPT-4, the process is applied to a finite set of action classes per dataset. This manageable scale allows for manual verification, so after the initial sub-labels are generated by the LLM, each is reviewed by a human annotator to ensure correctness and consistency. We should also note that we also experimented with open-source LLMs such as LLaMA and Mistral for this task, but they did not produce satisfactory results. We also note that we evaluated open-source LLMs, such as LLaMA and Mistral, for class label decomposition; however, they did not produce satisfactory results.

---

**Prompt to GPT-4:**
*What are the main entity concept and the main motion concept in the action class [c]? To answer, I have provided some examples for you. For instance: In the action class 'Holding a book': entity = 'book', motion = 'holding'. In the action class 'Baseball pitch': entity = 'baseball', motion = 'throwing'. In the action class 'Walking': entity = 'None', motion = 'walking'.*

---

Figure 4: Prompt used to extract dense action–entity and action–motion classes from the original action classes in the dataset. Note: [c] refers to the text description of the original action class.

Here [c] refers to the text description of the original action class.

## A.2 MORE IMPLEMENTATION DETAILS

To implement our approach, we adapt PAT (Sardari et al. (2023)) as the backbone. PAT consists of several Relative Positional Transformer (RPT) components operating at different scales. To implement the Temporal Modeling Block in the Action-Entity sub-network, one RPT with the full scale is used to process the entire video segment. In contrast, for the Temporal Modeling Block in the Action-motion sub-network, the entire PAT is used as the backbone. Similar to PAT, which uses four RPT components for four different temporal scales, we also use four RPT components. Each RPT component has four self-attention layers, and in the Action-motion sub-network, we incorporate the cross-attention mechanism in the third layer. The feature dimension in the RPT blocks is 512.

For the input features (*i.e.*, CLIP, I3D, and optical flow), and to ensure a fair comparison, we use the publicly released features provided by the authors of Dai et al. (2022a).

We conducted our experiments using PyTorch on an NVIDIA GeForce RTX 3090 GPU. Our model was trained with the Adam optimizer, starting with an initial learning rate of 0.0001. We used a batch size of 1 for 30 epochs, a batch size of 5 for 30 epochs, and a batch size of 3 for 40 epochs and for TSU, Charades, and MultiTHUMOS, respectively. The learning rate was reduced by a factor of 10 after every 25 epochs for TSU, after every 7 epochs for Charades, and after every 35 epochs for MultiTHUMOS. Note that the different training settings for these three datasets are due to their varying sizes.

## A.3 MORE DETAILS ON GENERALIZATION OF OUR PROPOSED OPTIMIZATION

Our proposed loss operates by aligning the learned video representations in the embedding space with the text features of all co-occurring classes present in the input video. In our approach, the

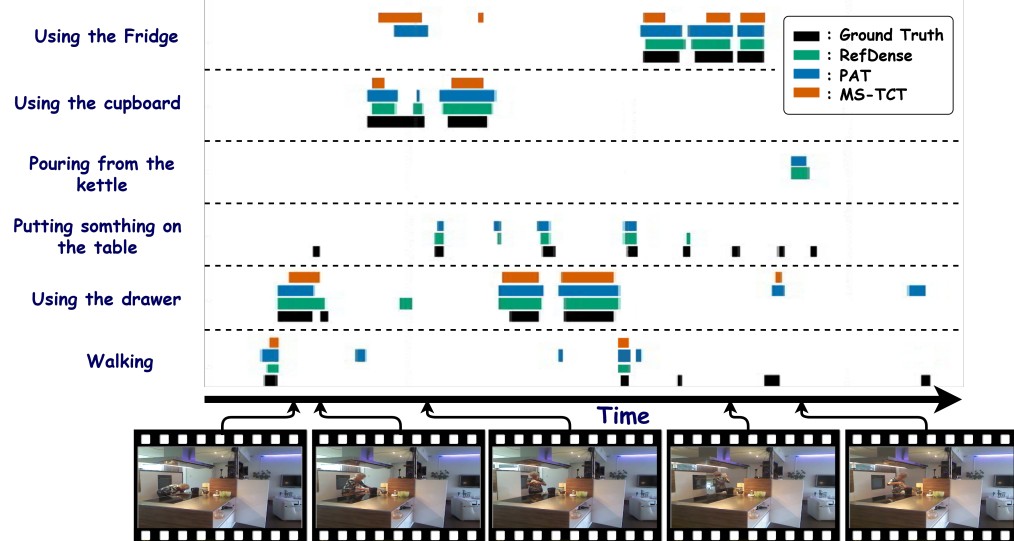

Figure 5: Qualitative comparison with previous approaches (PAT Sardari et al. (2023) and MS-TCT Dai et al. (2022a)) on a test video sample of TSU.

embedding space separately represents action-entities and action-motions. Therefore, when computing the loss in Eq. 7–9, we align the embedding space with the text features of all co-occurring action-entity and action-motion classes.

However, when applying our loss to the state-of-the-art methods, the entire embedding space is aligned with the text features of all co-occurring action classes instead. In fact, our loss is general and depends on the structure of the embedding space and the nature of the co-occurring concepts. The results reported in Table 6 are computed using the full embedding space and the corresponding action classes.

### A.4 MORE QUALITATIVE COMPARISON

In Fig. 5, we qualitatively compare our approach with the state-of-the-art methods PAT (Sardari et al. (2023)) and MS-TCT (Dai et al. (2022a)) on another test video sample. In this example, our method's predictions continue to align more closely with the ground-truth labels than those of the other methods. It also detects more action types than MS-TCT—for instance, MS-TCT fails to detect the action "Putting something on the table".

Furthermore, both PAT and MS-TCT struggle with class ambiguities in this example. For actions such as "Using the cupboard" and "Using the fridge", which exhibit motion overlap, act of "Using", they misclassify the latter and produce a false positive prediction. In contrast, our method correctly distinguishes between the two.

This test video also highlights noise in the ground-truth annotations. As shown in the last video frame image in Fig. 5, the action "Pouring from the kettle" is clearly occurring, and both PAT and RefDense successfully detect it, yet it is missing from the ground-truth labels.

### A.5 STATE-OF-THE-ART COMPARISON USING RGB + OPTICAL FLOW

In this section, we present the results of our proposed method with combined RGB and optical flow features, and compare them with state-of-the-art methods that also report results using these features. Table 7 shows the performance comparison with the standard metric, while Table 8 reports the comparison with action-conditional metrics. The results demonstrate that, consistent with our findings using only RGB features, our approach with combined features still outperforms other state-of-the-art methods. Specifically, it improves the standard mAP by an average of 0.9% across both datasets, and the action-conditional metrics by an average of 1.8% across all reported metrics.

Table 7: Dense action detection results using RGB + optical flow inputs, in terms of per-frame mAP. The best and the second best results are in **Bold** and underlined.

| Method | | mAP(%) | |
|---|---|---|---|
| | | Charades | MultiTHUMOS |
| MLAD (Tirupattur et al., 2021) CVPR | | 23.8 | 51.5 |
| CTRN (Dai et al., 2021a) BMVC | | 27.8 | 51.2 |
| **RefDense** | | **28.5** | **52.6** |
| | | (+0.7) | (+1.1) |

Table 8: Dense action detection results on Charades using RGB + optical flow inputs, evaluated based on the action-conditional metrics with cross-action dependencies over a temporal window of size $\delta$. The best and the second best results are in **Bold** and underlined.

| | $\delta = 0$ | | | | $\delta = 20$ | | | | Avg |
|---|---|---|---|---|---|---|---|---|---|
| | $mAP_{ac}$ | $F1_{ac}$ | $P_{ac}$ | $R_{ac}$ | $mAP_{ac}$ | $F1_{ac}$ | $P_{ac}$ | $R_{ac}$ | |
| MLAD (Tirupattur et al., 2021) | 29.0 | 8.9 | 19.4 | 7.3 | 35.7 | 10.5 | 18.9 | 8.9 | |
| CTRN (Dai et al., 2021a) | 29.7 | 11.9 | 23.9 | 8.0 | 36.8 | 12.9 | 27.1 | 9.1 | |
| MS-TCT (Dai et al., 2022a) | 30.7 | 19.5 | 26.3 | 15.5 | 37.6 | 22.1 | 27.6 | 18.4 | |
| PAT (Sardari et al., 2023) | 32.0 | 27.2 | **28.3** | 26.1 | 37.8 | 29.6 | 30.0 | 29.2 | |
| **RefDense** | **32.3** | **28.9** | 27.7 | **30.1** | **38.6** | **32.3** | **30.9** | **33.8** | |
| | (+0.3) | (+1.7) | (-0.6) | (+4.0) | (+0.8) | (+2.9) | (+0.9) | (+4.6) | (+1.8) |

## A.6 DETAILED IMPACT OF ACTION-ENTITY AND ACTION-MOTION DETECTION SUB-PROBLEMS

In Table 4, we study the impact of the sub-problems , dense action-entity detection and dense action-motion detection, on the performance of our proposed approach, RefDense. For a more detailed analysis, we report the effect of each sub-task individually in Table 9. The results demonstrate that eliminating each sub-task leads to a significant performance drop. Specifically, removing the dense action-motion detection sub-task decreases performance by 1.3% on $mAP_{ac}$, while removing the dense action-entity detection sub-task decreases performance by more than 1.8% across all metrics. Notably, the highest performance is achieved when both sub-problems are included.

Table 9: Ablation studies on the impact of sub-problems on the TSU dataset. The action conditional metrics are computed over a temporal window of size $\delta = 0$.

| Sub-task | | TSU | |
|---|---|---|---|
| | mAP | $mAP_{ac}$ | $F1_{ac}$ |
| Dense Action-Entity Detection | 42.9 | 31.9 | 29.9 |
| Dense Action-Motion Detection | 40.4 | 31.4 | 28.3 |
| Dense Action-Entity Detection, Dense Action-Motion Detection | **43.4** | **33.2** | **30.3** |

## A.7 MORE ANALYSIS ON THE IMPACT OF TASK DECOMPOSITION

When the network is optimized using the sub-tasks of dense entity and motion detection, it may naturally gain the benefit of learning foreground-focused representations. However, to demonstrate that the contribution of our decomposition strategy is not limited to foreground filtering, and that the network also benefits from learning higher-level action-entity and action-motion semantic concepts related to the actions, we provide the ablation studies in Table 10. In this table, we compare RefDense under three conditions: (i) using action-entity and action-motion sub-labels, (ii) using sub-labels redefined solely for foreground entity and motion detection, and (ii) removing all sub-labels entirely. For condition (i), the "foreground-only" variant, we redefine the sub-labels so that they no longer encode semantic distinctions between entities and motions. Instead, each sub-label simply indicates whether a frame contains any entity or motion foreground (label = 1) or is background (label = 0).

Table 10: Ablation studies on the impact of different sub-tasks on the TSU dataset. The action conditional metrics are computed over a temporal window of size $\delta = 0$.

| Network | sub-tasks | TSU | | |
| --- | --- | --- | --- | --- |
| | | mAP | mAP$_{ac}$ | F1$_{ac}$ |
| RefDense | No sub-task | 42.3 | 31.8 | 29.0 |
| RefDense | Dense Action-Entity Detection & Dense Action-Motion Detection | **43.4** | **33.2** | **30.3** |
| RefDense | Dense Foreground-Entity Detection & Dense Foreground-Motion Detection | 42.5 | 31.8 | 29.0 |

Table 11: Ablation studies on the performance of sub-networks on the TSU and Charades datasets. The action conditional metrics are computed over a temporal window of size $\delta = 0$.

| Sub-networks | TSU | | | Charades | | |
| --- | --- | --- | --- | --- | --- | --- |
| | mAP | mAP$_{ac}$ | F1$_{ac}$ | mAP | mAP$_{ac}$ | F1$_{ac}$ |
| Action-Entity | 35.7 | 26.6 | 24.3 | 53.4 | 58.1 | 54.1 |
| Action-Motion | 47.8 | 37.6 | 34.4 | 47.5 | 53.2 | 47.1 |
| RefDense | 42.3 | 31.8 | 29.0 | 34.1 | 38.6 | 33.1 |

The results in Table 10 show that this foreground-only supervision yields only a marginal improvement of 0.2% mAP, and provides no improvement on the action-conditional metrics. In contrast, the full semantic decomposition in condition (i) produces substantially larger gains (1.1% mAP, 1.4% mAP$_{ac}$, and 1.3% F1$_{ac}$, respectively). This confirms that the benefits of our approach cannot be explained by foreground detection alone, and that the model indeed leverages the semantic structure introduced by the action-entity and action-motion sub-tasks.

## A.8 PERFORMANCE OF ACTION-ENTITY AND ACTION-MOTION SUB-NETWORKS

Table 11 presents the performance of the Action-Entity and Action-Motion sub-networks on their respective tasks—dense action-entity detection and dense action-motion detection. Comparing their performance with that of RefDense shows that, consistent with the overall action-detection results, each sub-network performs well at detecting the specific semantic concepts it is designed to model. In Fig. 6, we also show the qualitative performance of the sub-networks for a video sample of the Charades dataset.

## A.9 IS WELL ESTABLISHED TWO-STREAM PARADIGM EFFECTIVE ENOUGH FOR DENSE ACTION DETECTION?

In Section 2, we stated that the well-established two-stream paradigm, when trained holistically using only action labels, cannot effectively learn the high-level semantic concepts essential for addressing class ambiguity. Thus, simply using different modalities or architectural designs is not sufficient; the network requires explicit guidance to learn these semantics. To study this claim, in this section we adapt RefDense to use different modalities for the Action-Entity and Action-Motion sub-networks (*i.e.*, RGB features and optical flow, respectively), following prior state-of-the-art works such as Simonyan & Zisserman (2014); Carreira & Zisserman (2017). We then train the adapted model under two settings: (i) the entire network is trained holistically using only dense action-detection labels for the main task, and (ii) the two streams are additionally trained using the decomposed semantic sub-labels for the dense action-entity and dense action-motion sub-tasks, respectively.

Our results in Table 12 show that explicitly teaching the sub-networks through these semantic sub-labels leads to a significant performance improvement (over 1.0% across all metrics). This demonstrates that the two-stream paradigm alone is not capable of acquiring these semantic concepts, and that semantic decomposition is crucial for enabling the model to learn the high-level structures necessary to resolve class ambiguity.

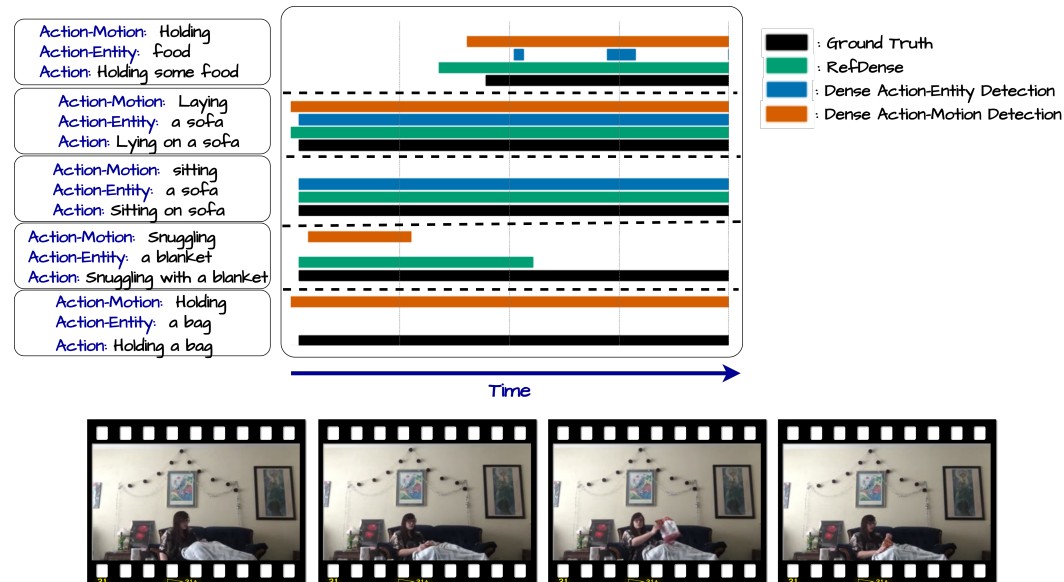

Figure 6: Qualitative performance of the sub-networks of RefDense on a video sample from the Charades dataset.

Table 12: Ablation studies on the impact of sub-tasks while using different modalities for Action-Entity and Action-Motion sub-networks on the Charades dataset. The action conditional metrics are computed over a temporal window of size $\delta = 0$.

| Networks | Input | Tasks | mAP | $mAP_{ac}$ | $F1_{ac}$ |
|---|---|---|---|---|---|
| RefDense | RGB & optical flow | Dense Action Detection | 33.0 | 38.1 | 33.1 |
| RefDense | RGB & optical flow | Dense Action Detection & Dense Action-Entity Detection & Dense Action-Motion Detection | 34.2 (+1.2) | 39.3 (+1.2) | 34.5 (+1.4) |

## A.10 LIMITATIONS

In our proposed approach, the decomposed sub- labels (dense action-entity and dense action-motion labels) preserve the temporal boundaries of the original dense action labels, as they are generated automatically to avoid manual annotation costs. However, for some samples, the sub-label boundaries may not fully overlap with the original action labels. Improving boundary accuracy could enhance performance. Our future work will also focus on refining these boundaries, but without relying on costly manual annotation.

