# OpenReview forum: "Reframing Dense Action Detection (RefDense): A New Perspective on Problem Solving and a Novel Optimization Strategy"
_ICLR.cc/2026/Conference — Submitted to ICLR 2026_

### Official Review · Reviewer_Nh85 · 2025-10-19

**Soundness:** 3
**Presentation:** 3
**Contribution:** 2
**Rating:** 4
**Confidence:** 4

**Summary:**

This paper addresses the task of temporal dense action detection, a fundamental and challenging problem in video understanding. The authors propose to decompose dense action labels into two components: action-entity and action-motion, to alleviate the difficulty of modeling overlapping and co-occurring actions. In addition, they employ a noisy contrastive learning objective to provide explicit supervision for co-occurring concepts. Experiments on three benchmark datasets show moderate performance gains compared to prior methods. While the paper is clear and well organized, the conceptual novelty is somewhat limited, and some claims are overstated. The decomposition into entity and motion branches aligns closely with well-established two-stream and relational modeling paradigms in video understanding. Furthermore, the method introduces additional network capacity, making it difficult to disentangle gains due to the proposed design from those due to the larger network.

**Strengths:**

-	This paper is well organized and easy to follow.
-	This paper targets a general and important task for video understanding, temporal dense action detection.
-	The decomposition of actions into entity and motion components is conceptually intuitive and may help address overlapping action scenarios.

**Weaknesses:**

-	The paper claims novelty in addressing simultaneous temporal and class overlaps, but this challenge has been widely recognized in earlier dense detection and multi-label video models.

-	The statement (L156–L157) suggesting that prior two-stream networks focus only on low-level spatiotemporal features because they are trained end-to-end lacks conceptual clarity and justification.

-	The two-stream design introduces increased model capacity, making performance comparisons with single-stream baselines potentially unfair. The ablation studies do not convincingly separate the effects of decomposition from additional parameters.

**Questions:**

The main technical concerns are outlined in the weaknesses section.

A minor question relates to the results. I noticed this manuscript was made public earlier this year, but the results in that version differ from those in the current one. Given that the overall methodology remains largely unchanged, what are the key differences?

---

> ### Author Response · Authors · 2025-11-16
> **Author's responces to the comment 1 of Reviewer Nh85**
>
> **First, we would like to thank the Reviewer Nh85 for taking time to reviewe our work and their valuable feedback.**
>
> **Reviewer's Comment:** The paper claims novelty in addressing simultaneous temporal and class overlaps, but this challenge has been widely recognized in earlier dense detection and multi-label video models.
>
> **Author's Response:** We agree that prior dense action detection methods may **implicitly** handle simultaneous temporal and class overlaps through their architectural design. However, our contribution lies in **explicitly** identifying the dual challenge of simultaneous temporal and class overlaps- an aspect that has not been **explicitly** explored in prior work —which opens new opportunities for future research. To the best of our knowledge, for the first time, we introduce this challnge. we additionally propose the first framework specifically designed to **explicitly** address this challenge. We have revised L104-105 of the manuscript to clarify this notion of **explicit** handling. If there exist earlier works that explicitly formulate and address this dual challenge, we would greatly appreciate the reviewer pointing them out so that we can properly acknowledge and discuss them.

---

> ### Author Response · Authors · 2025-11-16
> **Author's responces to the comment 2 of Reviewer Nh85**
>
> **Reviewer's Comment:** The statement (L156–L157) suggesting that prior two-stream networks focus only on low-level spatiotemporal features because they are trained end-to-end lacks conceptual clarity and justification.
>
> **Author's Response:** In two-stream networks, although spatial and temporal information are modeled using different modalities and/or architectures, the networks often fail to effectively learn high-level entity and motion semantics. This limitation arises because the networks are optimized end-to-end for the overall action-detection task, without **explicit** supervision guiding them to learn high-level semantics. In sparse-label scenarios—where at most one action occurs at each timestamp—end-to-end optimization may still encourage the streams to **implicitly** learn these semantics. For example, when a network observes a person taking a mug, the spatial stream may implicitly learn to focus on the entity (the person and the mug), while the temporal stream may implicitly capture the motion (the hand reaching for and lifting the mug).  However, in dense multi-label settings, where multiple actions can occur simultaneously and many classes share overlapping semantic components (action class ambiguities), even separate streams with end-to-end optimization struggle to effectively learn or disentangle these semantics. In contrast, our approach **explicitly** learns high-level semantic concepts within each stream by leveraging dense temporal action-entity and action-motion sub-label supervision, derived from the original action classes and annotations.
>
> We have revised the Related Work section (L153-161) to better reflect these points. To support this point, we also added ablation studies (Appendix A.9) in which we adapt Refenes to use different modalities (RGB and optical flows) as in two-stream paradigm. We then train the adapted model under two settings: (i) the entire network is trained holistically using only dense action-detection labels for the main task, and (ii) the stream streams are additionally trained using the decomposed semantic sub-labels for the dense action-entity and dense action-motion sub-tasks, respectively. Our results in this section show that explicitly teaching the subnetworks through these semantic sub-labels leads to a significant performance improvement (+1.2% on mAP, +1.2% on mAPac, and +1.4% on F1ac).

---

> ### Author Response · Authors · 2025-11-16
> **Author's responces to the comment 3 of Reviewer Nh85**
>
> **Reviewer's Comment:** The two-stream design introduces increased model capacity, making performance comparisons with single-stream baselines potentially unfair. The ablation studies do not convincingly separate the effects of decomposition from additional parameters.
>
> **Author's Response:** We would like to clarify that the **ablation studies in Table 5 of the original manuscrip are specifically designed to evaluate the impact of semantic decomposition while controlling for model capacity**. We use two state-of-the-art models, PAT and MS-TCT, as baselines and compare their performance under two settings: (i) training each model as a single unified network for dense action detection, and (ii) embedding them as Action-Entity and Action-Motion sub-networks within our proposed framework.
>
> To ensure a fair comparison, both the unified and dual-branch frameworks use the same total embedding dimensionality (i.e., each branch in the two-stream framework has half the embedding size of the unified network). This design actually makes the total number of parameters in dual-branch frameworks less than the unified framework because the transformer’s parameters scale roughly as D²: replacing a single weight matrix of size D_total² with two matrices of size (D_total/2)² results in total parameters of 2*(D_total/2)² = D_total² / 2 < D_total².
>
> As shown in Table 5, despite having fewer parameters, the proposed dual-branch framework significantly improves performance for both PAT and MS-TCT, achieving an average gain of over 2.0% across all metrics. This demonstrates that the performance improvement is due to semantic decomposition rather than increased model capacity.

---

> ### Author Response · Authors · 2025-11-16
> **Author's responces to the minor question of Reviewer Nh85**
>
> **Reviewer's Question:** I noticed this manuscript was made public earlier this year, but the results in that version differ from those in the current one. Given that the overall methodology remains largely unchanged, what are the key differences?
>
> **Author's Response:** The differences in our results arise from two main factors:
>
> (1) Unified modalities for fairness: In the previous version of the paper, the Action-Entity and Action-Motion subnetworks used different feature modalities. In the current version, to ensure a fair comparison with state-of-the-art methods, both streams use the same modality—either CLIP or I3D features.
>
> (2) Use of standardized input features: The earlier version relied on features extracted by ourselves. For fair and consistent comparison, in this version, we contacted the authors of MS-TCT and ANN papers to share the complete set of input features they used. So, the experiments in this version were performed with these features.

---

> > ### Comment · Reviewer_Nh85 · 2025-11-26
> >
> > I would like to appreciate the authors' great efforts and extensive responses. I would like to increase my initial score to 5, but this year, the metrics only contain 4 or 6 around the borderline. So it is hard to make such decision. As a result, I maintain my initial score.

---

> ### Author Response · Authors · 2025-11-26
> **Request for Clarification on Maintained Rating**
>
> Dear Reviewer Nh85,
>
> Thank you for your thoughtful follow-up and for acknowledging our efforts in addressing the concerns you previously raised.
>
> Since **we have carefully addressed all of the points mentioned in the earlier review, conducted extensive additional experiments, and revised the paper accordingly**, we would greatly appreciate it **if you could clarify the remaining reasons for retaining the original rating**. If there are any specific issues that still require further explanation, we would be more than happy to provide additional clarification.
>
> **Regarding the contributions of our work**, we would like to reiterate that our paper tackles one of the most challenging tasks in video understanding—dense action detection. As highlighted in the related work, **only a limited number of studies have addressed this problem in recent years (with the most recent one published at BMVC 2023), and our paper will benefit the community in this challengig task**.
>
> **Importantly, our paper is not incremental; rather, it introduces three novel and significant contributions to the field:**
>
> 1- We identify, for the first time, the challenge of simultaneous temporal and class overlaps in dense action detection, **an aspect not explicitly explored in prior work. We believe this recognition opens new opportunities for future research.**
>
> 2- To address this challenge, we propose a novel problem-solving perspective by decomposing the problem complexity for the network. **This design principle can also benefit other dense computer vision tasks, such as dense video captioning.**
>
> 3- We introduce an optimization strategy that explicitly leverages supervision on co-occurring concepts during training, **providing a new direction for improving dense prediction models.**
>
> We hope this clarification is helpful, and we would be grateful for any additional guidance you can provide regarding the remaining concerns underlying the current score.

---

### Official Review · Reviewer_QKDi · 2025-10-25

**Soundness:** 2
**Presentation:** 2
**Contribution:** 2
**Rating:** 4
**Confidence:** 4

**Summary:**

This paper introduces RefDense, a framework designed for dense action detection that addresses the challenges of temporal and class overlaps through problem decomposition. The approach consists of two key components: first, actions are decomposed into entity and motion components, with dedicated sub-networks tasked to detect each, thereby simplifying individual learning objectives. Second, a contrastive co-occurrence loss leveraging language guidance is proposed to explicitly capture relationships among frequently co-occurring actions, overcoming the limitation of standard binary cross-entropy loss that treats classes independently. The method is evaluated on the TSU, Charades, and MultiTHUMOS datasets.

**Strengths:**

1. The paper is clearly written and easy to follow.
2. The proposed method demonstrates performance gains ranging from 0.4% to 2.1% across the benchmark datasets.

**Weaknesses:**

1. The idea of decomposing actions into entity and motion components bears resemblance to established paradigms like two-stream networks and several recent works.  [1] Dual detrs for multi-label temporal action detection, CVPR 2024.
[2] Decomposed cross-modal distillation for rgb-based temporal action detection, CVPR 2023.

2. The construction of labels for the sub-networks, specifically the "action-entity" labels, may be problematic. In untrimmed videos, certain entities (e.g., "hammer" in the provided example) might be present throughout the entire video duration, even when the corresponding action is not being performed. This could lead to ambiguous and noisy supervision for the Action-Entity sub-network. The authors should address this potential issue and justify the robustness of their labeling process.

3. The experimental comparisons appear to be limited to other dense action detection methods. To better position the work, it would be valuable to include comparisons with recent state-of-the-art methods in temporal action localization on the same datasets, which would provide a broader perspective on its performance.

4. The ablation studies could be more comprehensive. Key questions remain unanswered: What is the performance of each sub-network (Action-Entity and Action-Motion) when trained and evaluated independently? Is the observed performance gain primarily due to the increased network capacity (using two sub-networks) or the core idea of decomposition? A controlled experiment, for instance, comparing against a single network of comparable parameters, would help isolate the true source of improvement.

5. The figures could be improved for clarity: Figure 1 would benefit from concrete examples of actions (e.g., "pour water") to more directly illustrate the concepts of entity and motion decomposition. There is a typo in Figure 2; the second sub-network is currently labeled "Action-Entity" but should presumably be "Action-Motion."

6. There is a confusing use of the symbol tau in the manuscript. It is used to represent the temperature coefficient in Equation 8 but denotes a window size in Table 2. To avoid confusion for the reader, it is strongly recommended to use distinct symbols for these different parameters.

**Questions:**

1. What is the fundamental conceptual or technical advancement of your decomposition framework compared to these existing approaches?

2. How does the Action-Entity sub-network distinguish between an entity being merely present versus being actively involved in an action? Could you provide an analysis or examples from the validation set showing that the entity labels are temporally precise and not overly noisy?

3. How would your method, RefDense, perform against these recent temporal localization models in terms of average precision?

4. What is the standalone performance (e.g., on the decomposed task) of the Action-Entity and Action-Motion sub-networks?

5. Is the performance improvement primarily due to the increased model capacity from having two sub-networks? Have you conducted a controlled experiment comparing RefDense against a single, larger network with a comparable number of parameters?

---

> ### Author Response · Authors · 2025-11-16
> **Author's responces to the comment 1 and question 1 of Reviewer QKDi**
>
> **First, we would like to thank the Reviewer QKDi for taking time to reviewe our work and their valuable feedback.**
>
> **Reviewer's Comment and Question:** The idea of decomposing actions into entity and motion components bears resemblance to established paradigms like two-stream networks and several recent works. [1] Dual detrs for multi-label temporal action detection, CVPR 2024. [2] Decomposed cross-modal distillation for rgb-based temporal action detection, CVPR 2023. What is the fundamental conceptual or technical advancement of your decomposition framework compared to these existing approaches?
>
> **Author's Response:** Our work differs from existing two-stream networks in its method of learning action-entity and action-motion semantics, even though both approaches employ a two-stream architecture.
>
> In well-established two-stream networks, although separate streams are used for extracting spatial and temporal features, the entire framework is optimized **holistically using only the main action labels**. In contrast, in our approach, in addition to the primary action labels, the streams are optimized separately using **action-entity and action-motion sub-label supervision**, which are derived from the original action labels. This explicit sub-labeling is the key difference, enabling the effective learning of high-level entity and motion semantics within each stream in the dense labeld senario.
>
> In sparse-label scenarios—where at most one action occurs at each timestamp—holistic optimization using action labels may still encourage streams to **implicitly** learn these semantics. For instance, when a network observes a person taking a mug, the spatial stream may implicitly focus on the entity (the person and the mug), while the temporal stream may implicitly capture the motion (the hand reaching for and lifting the mug). However, in dense multi-label settings, where multiple actions can occur simultaneously and many classes share overlapping semantic components (leading to action class ambiguities), even separate streams with end-to-end action optimization **struggle to effectively learn or disentangle these semantics**. Our approach resolves this by **explicitly learning high-level semantic concepts** within each stream by leveraging dense temporal action-entity and action-motion sub-label supervision, derived directly from the original action classes and annotations.
>
> We also highlight that this conceptual paradigm has already discussed in detail in the Related Work section, L153-161, and in the revised paper, we have also refined it to better reflect the differences. Furthermore, we have added ablation studies (Appendix A.9) in the revised aper in which we adapt Refenes to use different modalities (RGB and optical flows) as in two-stream paradigm. We then train the adapted model under two settings: (i) the entire network is trained holistically using only dense action-detection labels for the main task, and (ii) the stream streams are additionally trained using the decomposed semantic sub-labels for the dense action-entity and dense action-motion sub-tasks, respectively. Our results in this section show that explicitly teaching the subnetworks through these semantic sub-labels leads to a significant performance improvement (+1.2% on mAP, +1.2% on mAPac, and +1.4% on F1ac).
>
> **Technical Distinction from Specific Works**
>
> **1. DualDETR (CVPR 2024)**: Regarding "[1] Dual DETRs for multi-label temporal action detection, CVPR 2024," while this is also a two-stream network, its goal differs fundamentally from ours. In DualDETR, both streams are designed to solve the same action detection task, with each stream focusing on learning different temporal granularities. In contrast, in our approach, our streams focus on different sub-tasks, aiming to learn distinct semantic concepts (action-entity and action-motion).
>
> **2. Decomposed Cross-Modal Distillation (CVPR 2023)**: Regarding this paper, as they follow the well-established two-stream paradigm, and their approach difference with our work has been explained above. Specifically, in this paper, the term "decomposition" is used for knowledge transfer between modalities (cross-modal distillation between RGB and Optical flows) to improve efficiency by reducing the inference to only one stream. By contrast, our method decomposes semantic sub-concepts (entity and motion) within the same modality, aiming to resolve semantic ambiguity in dense action detection rather than improving cross-modal efficiency.

---

> ### Author Response · Authors · 2025-11-16
> **Author's responces to the comment 2 and question 2 of Reviewer QKDi**
>
> **Reviewer's Comment and Question:** How does the Action-Entity sub-network distinguish between an entity being merely present versus being actively involved in an action? The construction of labels for the sub-networks, specifically the "action-entity" labels, may be problematic. In untrimmed videos, certain entities (e.g., "hammer" in the provided example) might be present throughout the entire video duration, even when the corresponding action is not being performed. This could lead to ambiguous and noisy supervision for the Action-Entity sub-network. The authors should address this potential issue and justify the robustness of their labeling process.
>
> **Author's Response:** As explained and formulated in lines 252–269 of the original manuscript, the sub-label supervision for both the Action-Entity and Action-Motion streams is derived directly from the temporal action annotations.  Even if an entity (e.g., a "hammer") appears throughout the entire video, it is labeled as present only during the temporal span of the corresponding action. (e.g., "hammering"). This is because the entity sub-labels are also temporal (i.e., they are defined on a frame-by-frame basis, not video-wide). This process ensures that the supervision signal is temporally precise and does not introduce ambiguity and noise in the Action-Entity stream's training. Consequently, the Action-Entity sub-network learns to focus on entities in the context of the actions they actively participate in, rather than merely their static presence in the scene.
>
> **Reviewer's  Question:** Could you provide an analysis or examples from the validation set showing that the entity labels are temporally precise and not overly noisy?
>
> **Author's Response:** As the reviewer requested, we have conducted a quantitative analysis of the sub-task performance for the Action-Entity and Action-Motion sub-networks. The results are included in the Appendix of the revised manuscript, Section A.8, Table 11. We would like to highlight that the performance of these sub-networks, when compared with RefDense, shows that—consistent with the overall action-detection results—each sub-network effectively detects the specific semantic concepts it is designed to model.

---

> ### Author Response · Authors · 2025-11-16
> **Author's responces to the comment 3 and question 3 of Reviewer QKDi**
>
> **Reviewer's Comment:** The experimental comparisons appear to be limited to other dense action detection methods. To better position the work, it would be valuable to include comparisons with recent state-of-the-art methods in temporal action localization on the same datasets, which would provide a broader perspective on its performance.
>
> **Reviewer Question:** How would your method, RefDense, perform against these recent temporal localization models in terms of average precision?
>
> **Author's Response:** We would like to clarify and emphasize that the scope of our paper is dense multi-label action detection, which is fundamentally different from single-label (sparse) temporal action localization (TAL). **Sparse-label and dense-label action detection represent two distinct problem domains, differing in their challenges, evaluation metrics, and benchmark datasets**. Our method is specifically designed to address the challenges of multi-label, overlapping actions, and applying it to the sparse single-action setting would not be meaningful or fair as method designed for dense multi-label detection may not be suitable for sparse-label TAL tasks, and vice versa.
>
> We would also like to highlight that our method is evaluated following the established SOTA protocol for dense multi-label action detection (e.g., PAT, MS-TCT, MLAD, ANN). We conduct extensive experiments on all existing benchmark datasets in this area and report all standard evaluation metrics. Furthermore, as detailed in lines 376–403, unlike many prior works that primarily report only standard mAP, we additionally provide comprehensive action-conditional metric results, which offer a more informative assessment of model behavior in dense multi-label scenarios.

---

> ### Author Response · Authors · 2025-11-16
> **Author's responces to the comment 4 of Reviewer QKDi**
>
> **Reviewer's Comment:** The ablation studies could be more comprehensive. Key questions remain unanswered: What is the performance of each sub-network (Action-Entity and Action-Motion) when trained and evaluated independently? Is the observed performance gain primarily due to the increased network capacity (using two sub-networks) or the core idea of decomposition? A controlled experiment, for instance, comparing against a single network of comparable parameters, would help isolate the true source of improvement.
>
> **Author's Response:** We would like to clarify that the **ablation studies in Table 5 of the original manuscrip are specifically designed to evaluate the impact of semantic decomposition while controlling for model capacity**. We use two state-of-the-art models, PAT and MS-TCT, as baselines and compare their performance under two settings: (i) training each model as a single unified network for dense action detection, and (ii) embedding them as Action-Entity and Action-Motion sub-networks within our proposed framework.
>
> To ensure a fair comparison, both the unified and dual-branch frameworks use the same total embedding dimensionality (i.e., each branch in the two-stream framework has half the embedding size of the unified network). This design actually makes the total number of parameters in dual-branch frameworks less than the unified framework because the transformer’s parameters scale roughly as D²: replacing a single weight matrix of size D_total² with two matrices of size (D_total/2)² results in total parameters of 2*(D_total/2)² = D_total² / 2 < D_total².
>
> As shown in Table 5, despite having fewer parameters, the proposed dual-branch framework significantly improves performance for both PAT and MS-TCT, achieving an average gain of over 2.0% across all metrics. This demonstrates that the performance improvement is due to semantic decomposition rather than increased model capacity.

---

> ### Author Response · Authors · 2025-11-16
> **Author's responces to the question 4 of Reviewer QKDi**
>
> **Reviewer's Question:** What is the standalone performance (e.g., on the decomposed task) of the Action-Entity and Action-Motion sub-networks?
>
> **Author's Response:** The performance of the sub-networks on the decomposed tasks have been added to the revised manuscript. Please see the Appendix, Section A.8, and Table 11. We would like to highlight that Table 11 presents the performance of the Action-Entity and Action-Motion sub-networks on their respective tasks—dense action-entity detection and dense action-motion detection. Comparing their performance with that of RefDense shows that, consistent with the overall action-detection results, each sub-network performs well at detecting the specific semantic concepts it is designed to model. We additionally also obtained the qualitative results of Action-Entity and Action-Motion sub-networks on Figure 6 of the revised manuscript.

---

> ### Author Response · Authors · 2025-11-16
> **Author's responces to the comment 5 of Reviewer QKDi**
>
> **Reviewer's Comment:** The figures could be improved for clarity: Figure 1 would benefit from concrete examples of actions (e.g., "pour water") to more directly illustrate the concepts of entity and motion decomposition.
>
>  **Author's Response:** We appreciate the reviewer’s suggestion regarding Figure 1. However, we would be grateful if the reviewer could give us more details on how to illustrate those concepts specifically with the limited space we have for that figure.
>
> **Reviewer's Comment:** There is a typo in Figure 2; the second sub-network is currently labeled "Action-Entity" but should presumably be "Action-Motion."
>
> **Author's Response:** We would like to thank the reviewer for pointing this. Now the Figure is corrected in the revised paper.

---

> ### Author Response · Authors · 2025-11-16
> **Author's responces to the comment 6 of Reviewer QKDi**
>
> **Reviewer's Comment:** There is a confusing use of the symbol tau in the manuscript. It is used to represent the temperature coefficient in Equation 8 but denotes a window size in Table 2. To avoid confusion for the reader, it is strongly recommended to use distinct symbols for these different parameters.
>
> **Author's Response:** We would like to thank the reviewer for pointing out this issue. In the revised version of the manuscript, we have replaced the symbol tau used for the window size with delta throughout the paper to avoid confusion with the temperature coefficient.

---

> ### Author Response · Authors · 2025-11-16
> **Author's responces to the question 5 of Reviewer QKDi**
>
> **Reviewer's Comment:** Is the performance improvement primarily due to the increased model capacity from having two sub-networks? Have you conducted a controlled experiment comparing RefDense against a single, larger network with a comparable number of parameters?
>
> **Author's Response:** Please see our response to the comment 4.

---

> ### Author Response · Authors · 2025-11-28
> **Our paper's contributions:**
>
> **We would like to higlight that our paper ackles one of the most challenging tasks in video understanding—dense action detection**. As highlighted in the related work, **only a limited number of studies have addressed this problem in recent years (with the most recent one published at BMVC 2023), and our paper will benefit the community in this challengig task**.
>
> **Importantly, our paper is not incremental; rather, it introduces three novel and significant contributions to the field:**
>
> 1- We identify, for the first time, the challenge of simultaneous temporal and class overlaps in dense action detection, **an aspect not explicitly explored in prior work. We believe this recognition opens new opportunities for future research.**
>
> 2- To address this challenge, we propose a novel problem-solving perspective by decomposing the problem complexity for the network. **This design principle can also benefit other dense computer vision tasks, such as dense video captioning.**
>
> 3- We introduce an optimization strategy that explicitly leverages supervision on co-occurring concepts during training, **providing a new direction for improving dense prediction models.**
>
> We hope this clarification is helpful, and we would be grateful for any additional guidance you can provide regarding the remaining concerns underlying the current score.

---

### Official Review · Reviewer_DHHi · 2025-10-29

**Soundness:** 2
**Presentation:** 2
**Contribution:** 1
**Rating:** 2
**Confidence:** 4

**Summary:**

The manuscript suffers from critical flaws in methodological transparency, experimental rigor, and practical relevance that cannot be addressed through minor revisions. The lack of reproducible details for label decomposition and L_CoLV, confounded generalization experiments, and incomplete engagement with related work undermine the validity of the claimed contributions. To be reconsidered, the authors would need to: (1) fully specify all methodological details (e.g., L_CoLV formulation, GPT-4 prompts), (2) conduct ablation studies to isolate the impact of key components (e.g., parameter count vs. decomposition), (3) validate performance across more diverse qualitative examples, and (4) address practical constraints like computational efficiency and LLM accessibility.

**Strengths:**

This paper proposed a strategy of decomposing the task into detecting temporally dense but unambiguous components underlying the action classes, and assigning these sub-problems to distinct sub-networks

**Weaknesses:**

1. The core contributions of RefDense—action label decomposition via GPT-4 and Contrastive Co-occurrence Language-Video Loss (L_CoLV)—lack sufficient detail to support reproducibility and validity
2. The experimental evaluations, while extensive, suffer from biases, unaddressed confounders, and incomplete reporting that undermine the credibility of the claimed performance gains. For example, Confounded Generalization Experiments: When embedding PAT and MS-TCT into the RefDense framework (Table 5), the paper reduces the parameter count of each sub-network (e.g., PAT from 270M to 144M) while claiming "total embedding dimensionality is the same."
3. Inconsistent SOTA Benchmarking: Many SOTA comparisons rely on re-run results (marked †) using the authors’ own code, but fail to validate that experimental conditions (e.g., optimizer hyperparameters, training epochs, data augmentation) match the original papers.
4. Insufficient Discussion of Limitations and Practicality. The paper does not evaluate decomposition performance with open-source LLMs (e.g., LLaMA-3, Mistral) to assess accessibility. Besides, this paper does not discuss the scalability of LLM-based label generation for larger datasets (e.g., beyond 10k videos in Charades).
5. The related work section fails to engage with recent or relevant literature, leading to an inaccurate positioning of RefDense’s novelty. For example, it oversimplifies Two-Stream Networks and neglects Vision-Language action detection precedents:

**Questions:**

1. The core contributions of RefDense—action label decomposition via GPT-4 and Contrastive Co-occurrence Language-Video Loss (L_CoLV)—lack sufficient detail to support reproducibility and validity
2. The experimental evaluations, while extensive, suffer from biases, unaddressed confounders, and incomplete reporting that undermine the credibility of the claimed performance gains. For example, Confounded Generalization Experiments: When embedding PAT and MS-TCT into the RefDense framework (Table 5), the paper reduces the parameter count of each sub-network (e.g., PAT from 270M to 144M) while claiming "total embedding dimensionality is the same."
3. Inconsistent SOTA Benchmarking: Many SOTA comparisons rely on re-run results (marked †) using the authors’ own code, but fail to validate that experimental conditions (e.g., optimizer hyperparameters, training epochs, data augmentation) match the original papers.
4. Insufficient Discussion of Limitations and Practicality. The paper does not evaluate decomposition performance with open-source LLMs (e.g., LLaMA-3, Mistral) to assess accessibility. Besides, this paper does not discuss the scalability of LLM-based label generation for larger datasets (e.g., beyond 10k videos in Charades).
5. The related work section fails to engage with recent or relevant literature, leading to an inaccurate positioning of RefDense’s novelty. For example, it oversimplifies Two-Stream Networks and neglects Vision-Language action detection precedents:

---

> ### Author Response · Authors · 2025-11-22
> **Author's responces to the comment 1 and question 1 of Reviewer DHHi**
>
> **Reviewer's Comment and Question:** The core contributions of RefDense—action label decomposition via GPT-4 and Contrastive Co-occurrence Language-Video Loss (L_CoLV)—lack sufficient detail to support reproducibility and validity.
>
> **Author's Response:** We disagree with the reviewer. First, both the label decomposition and the proposed L_{CoLV} loss are extensively described and formalized in the manuscript (lines L254–272 for label decomposition and lines L296–318 for L_{CoLV}). The L_{CoLV} loss is an adaptation of the well-established Noise Contrastive Estimation (NCE) loss and can be implemented directly using the equations provided in lines L293–315. For L_{CoLV}, we also included an illustrative figure to aid understanding (Figure 2(b)), and the GPT prompt used for the label decomposition is provided in the appendix (Figure 4). Therefore, these components can be readily reproduced using the provided formulations. Furthermore, as stated in the Reproducibility Statement section, both the sub-labels and the code will be released upon paper publication.
>
> **If the reviewer identifies any specific part of the label decomposition or L_{CoLV} that remains unclear or difficult to reproduce, we would appreciate detailed feedback so that we can address or clarify it accordingly.**

---

> ### Author Response · Authors · 2025-11-22
> **Author's responces to the comment 2 and question 2 of Reviewer DHHi**
>
> **Reviewer's Comment and Question:** The experimental evaluations, while extensive, suffer from biases, unaddressed confounders, and incomplete reporting that undermine the credibility of the claimed performance gains. For example, Confounded Generalization Experiments: When embedding PAT and MS-TCT into the RefDense framework (Table 5), the paper reduces the parameter count of each sub-network (e.g., PAT from 270M to 144M) while claiming "total embedding dimensionality is the same."
>
> **Author's Response:** We believe our ablation studies were designed to thoroughly and fairly isolate the contribution of each component, and in the revised manuscript we have added additional experiments to further strengthen this analysis. **If there are particular cases where the reviewer believes a bias or confounder remains unaddressed, we would appreciate if they could point them out so that we can clarify or incorporate the necessary controls.**
>
> Specifically, about the reviewer's comment on Table 5, we would like to clarify that embedding dimensionality is distinct from the total number of parameters. Our claim regarding the embedding size is correct, and the total number of parameters aligns with these dimensions. In a transformer, if we replace a single layer of size D with two layers of size D/2, the number of parameters in each layer is reduced because transformer parameters scale roughly as D^2. Therefore, the total parameter count becomes 2.(D/2)^2 = (D^2)/2, which is smaller than the original single-layer transformer of size D.
>
> Importantly, this reduction in parameters applies only to our approach, while the PAT and MS-TCT baselines retain their original capacity. Despite using fewer parameters by our method in comparison to PAT and MS-TCT baselines, our method significantly improves performance, with an average gain of over 2% across all metrics. This demonstrates that the observed improvement is attributable to semantic decomposition rather than increased model capacity, making the comparison fair and valid.

---

> ### Author Response · Authors · 2025-11-22
> **Author's responces to the comment 3 and question 3 of Reviewer DHHi**
>
> **Reviewer's Comment and Question:** Inconsistent SOTA Benchmarking: Many SOTA comparisons rely on re-run results (marked †) using the authors’ own code, but fail to validate that experimental conditions (e.g., optimizer hyperparameters, training epochs, data augmentation) match the original papers.
>
> **Author's Response:** The reviewer raises concerns about SOTA comparisons. Based on prior SOTA works (e.g., MS-TCT, MLAD, and ANN), the standard protocol in dense action detection is to (1) use the same input features, (2) typically adopt a segment length of  T = 8, and (3) adjust training hyperparameters (e.g., epoch number, loss coefficients) as needed due to differences in network architecture. We would like to emphasis that we strictly follow this standard protocol:
>
> 1- Input features: We use the same features released by the original works and have revised the manuscript to clearly reflect this (see green lines in Appendix, Section A.2).
>
> 2- Segment length: We use T=8 (see line 354 of the paper).
>
> 3- Training hyperparameters: Consistent with prior SOTA studies (e.g., MS-TCT, MLAD, ANN) that adapt learning rates, training epochs, and other hyperparameters to their own architectures, we also adopt our own training hyperparameters.

---

> ### Author Response · Authors · 2025-11-22
> **Author's responces to the comment 4 and question 4 of Reviewer DHHi**
>
> **Reviewer's Comment and Question:**  Insufficient Discussion of Limitations and Practicality.
>
> **Author's Response:**  **We would like to highlight that the method’s limitations are discussed in the original manuscript (Appendix, Section A.10).** Regarding practicality, the introduction provides an extensive discussion of our method and its motivation. As stated in the Contributions section (lines L103–113), we identify, for the first time, the challenge of simultaneous temporal and class overlaps in dense action detection, which opens new opportunities for future research. Our novel problem-solving perspective—decomposing the task into sub-tasks—can also benefit other dense computer vision tasks, such as dense captioning. Additionally, we introduce an optimization strategy that explicitly leverages supervision on co-occurring concepts during training, which can improve the performance of existing and future dense action detection networks.
>
> **Reviewer's Comment and Question:** The paper does not evaluate decomposition performance with open-source LLMs (e.g., LLaMA-3, Mistral) to assess accessibility. Besides, this paper does not discuss the scalability of LLM-based label generation for larger datasets (e.g., beyond 10k videos in Charades).
>
> **Author's Response:**  First, we would like to clarify that in our work **the LLM is not used to generate labels for individual video samples**. Instead, it is used offline to decompose only the action class list of the dataset, which is very limited (e.g., Charades has 156 classes, MultiTHUMOS has 56 classes). Using these decomposed action class lists along with the ground truth labels, the final action-entity and action-motion labels for each video are generated using the formulations in lines L254–271.
>
> Regarding open-source LLMs such as LLaMA-3 and Mistral, we attempted to use them for class label decomposition, but they did not produce satisfactory results. This clarification has been added in line L611-613 of the revised manuscript.
>
> Concerning scalability, **we repeat that that sub-labels are not generated per video by the LLM**. Instead, the LLM is used offline to decompose only the action class list of dataset. Since the action class list is independent of dataset size and always limited, our method easily scales to larger datasets, including those beyond 10k videos.

---

> ### Author Response · Authors · 2025-11-22
> **Author's responces to the comment 5 and question 5 of Reviewer DHHi**
>
> **Reviewer's Comment and Question:** The related work section fails to engage with recent or relevant literature, leading to an inaccurate positioning of RefDense’s novelty. For example, it oversimplifies Two-Stream Networks and neglects Vision-Language action detection precedents:
>
> **Author's Response:** We have extensively included all the most relevant works in our Related Work section. **If there are specific relevant works that we have missed, we would greatly appreciate the reviewer pointing them out so that we can discuss or include them.**
>
> Regarding two-stream networks, we have revised the Related Work section (lines L153–161) to better reflect the differences between traditional two-stream networks and our approach (see green lines in the revised manuscript). We have also added some ablation studies to support this in Section A.9 of Apendinx.
>
> Regarding Vision-Language approaches, we would like to clarify that our method is not a multi-modal (vision-language) action detection approach. Rather, it operates on a single visual modality, and language is used solely to guide the method’s optimization. We had included a Vision-Language action detection sub-section in the original manuscrip to cover relevant prior work in this area.

---

> > ### Comment · Reviewer_DHHi · 2025-11-25
> >
> > I acknowledge the rebuttal and retain my rating

---

> > > ### Author Response · Authors · 2025-11-27
> > >
> > > Dear Reviewer DHHi,
> > >
> > > Thank you for follow-up in addressing the concerns you previously raised.
> > >
> > > **Since we have carefully addressed all of the points mentioned in the earlier review**, we would greatly appreciate it if you could clarify the remaining reasons for retaining the original rating. If there are any specific issues that still require further explanation, we would be more than happy to provide additional clarification.

---

### Official Review · Reviewer_Nbui · 2025-10-29

**Soundness:** 3
**Presentation:** 3
**Contribution:** 2
**Rating:** 4
**Confidence:** 5

**Summary:**

This paper tackles the challenges of dense action detection, specifically temporal and class ambiguity, by proposing a decomposed approach. The method breaks down ambiguous actions into unambiguous temporal components, assigning them to specialized sub-networks to simplify temporal overlap resolution. Furthermore, it introduces a language-guided contrastive loss to explicitly model the relationships between co-occurring actions, overcoming the limitations of independent class treatment in standard binary cross-entropy. The approach demonstrates superior performance, achieving substantial gains on TSU, Charades, and MultiTHUMOS benchmarks.

**Strengths:**

+ This paper decomposes the complex problem of dense action detection into simpler sub-tasks of detecting unambiguous temporal components, allowing specialized sub-networks to handle temporal overlaps more effectively.
+ The method demonstrates superior and substantial performance improvements over state-of-the-art methods across multiple challenging benchmark datasets.

**Weaknesses:**

- The performance gain might be better explained by the sub-networks specializing in foreground entities and actions. This specialization reduces the impact of the background after feature concatenation, which is a perspective that diverges from the authors' stated motivation.
- Missing visualization and quantitative results of two sub-network. The qualitative result comparison among the predicted action-entity, action-motion and the final detection result can help readers understand the reasons for the effectiveness.
- The performance improvements shown in Table 3 and Table 5 are incorrect. Please recheck these tables.

**Questions:**

None

---

> ### Author Response · Authors · 2025-11-16
> **Author's responces to the comment 1 of Reviewer Nbui**
>
> **First, we would like to thank the Reviewer Nbui for taking time to reviewe our work and their valuable feedback.**
>
> **Reviewer's Comment:** The performance gain might be better explained by the sub-networks specializing in foreground entities and actions. This specialization reduces the impact of the background after feature concatenation, which is a perspective that diverges from the authors' stated motivation.
>
> **Author's Response:** We agree with the reviewer that when the network is optimized using the sub-tasks of dense entity and motion detection, it may naturally gain the benefit of learning foreground representations. However, to demonstrate that the contribution of our decomposition strategy is not limited to foreground filtering, and that the network also benefits from learning higher-level action-entity and action-motion semantic concepts related to the actions, we provide the ablation studies in Table 10 of revised manuscript (please see Appendix, section A.7). In this table, we compare RefDense under three conditions: (i) using action-entity and action-motion sub-labels, (ii) using sub-labels redefined solely for foreground entity and motion detection, and (ii) removing all sub-labels entirely. For condition (ii), the “foreground-only” variant, we redefine the sub-labels so that they no longer encode semantic distinctions between entities and motions. Instead, each sub-label simply indicates whether a frame contains any entity or motion foreground (label = 1) or is background (label = 0).
>
> The results in Table 10 show that the foreground-only supervision yields only a marginal improvement of 0.2% mAP, and provides no improvement on the action-conditional metrics. This shows that with optimizing the network through the action labels, the network itself learns about the forground concepts at the same time, and the forground labels does not add extra supervion. In contrast, the full semantic decomposition in condition (i) produces substantially larger gains (1.1% mAP, 1.4% mAPac, and 1.3% F1ac, respectively). This confirms that the benefits of our approach cannot be explained by foreground detection alone, and that the model indeed leverages the semantic structure introduced by the action-entity and action-motion sub-tasks.
>
> We would like to thank the reviewer for pointing out this concern, as it motivated us to conduct this experiment, and the resulting findings further strengthen the contribution of our proposed idea.

---

> ### Author Response · Authors · 2025-11-16
> **Author's responces to the comment 2 of Reviewer Nbui**
>
> **Reviewer's Comment:** Missing visualization and quantitative results of two sub-network. The qualitative result comparison among the predicted action-entity, action-motion and the final detection result can help readers understand the reasons for the effectiveness.
>
> **Author's Response:** The quantitative results of the sub-networks have been added to the revised manuscript. Please see the Appendix, Section A.8, and Table 11. Table 11 presents the performance of the Action-Entity and Action-Motion sub-networks on their respective tasks—dense action-entity detection and dense action-motion detection. Comparing their performance with that of RefDense shows that, consistent with the overall action-detection results, each sub-network performs well at detecting the specific semantic concepts it is designed to model.
>
> We have also obtained the qualitative performance of the sub-networks of RefDense, please see the Figure 6 in the Appendix.

---

> ### Author Response · Authors · 2025-11-16
> **Author's responces to the comment 3 of Reviewer Nbui**
>
> **Reviewer's Comment:** The performance improvements shown in Table 3 and Table 5 are incorrect. Please recheck these tables.
>
> **Author's Response to Comment 3:** We would like to thank the reviewer for pointing this. Now the tables are updated with the correct numbers.

---

> ### Author Response · Authors · 2025-11-28
> **Our paper's contributions:**
>
> **We would like to higlight that our paper ackles one of the most challenging tasks in video understanding—dense action detection**. As highlighted in the related work, **only a limited number of studies have addressed this problem in recent years (with the most recent one published at BMVC 2023), and our paper will benefit the community in this challengig task**.
>
> **Importantly, our paper is not incremental; rather, it introduces three novel and significant contributions to the field:**
>
> 1- We identify, for the first time, the challenge of simultaneous temporal and class overlaps in dense action detection, **an aspect not explicitly explored in prior work. We believe this recognition opens new opportunities for future research.**
>
> 2- To address this challenge, we propose a novel problem-solving perspective by decomposing the problem complexity for the network. **This design principle can also benefit other dense computer vision tasks, such as dense video captioning.**
>
> 3- We introduce an optimization strategy that explicitly leverages supervision on co-occurring concepts during training, **providing a new direction for improving dense prediction models.**
>
> We hope this clarification is helpful, and we would be grateful for any additional guidance you can provide regarding the remaining concerns underlying the current score.

---

### Comment · Area_Chair_SkgZ · 2025-11-26
**Discussion**

Dear Reviewers,

Thank you for your efforts in reviewing this paper. The authors have submitted their responses; please take them into account when making your decision and acknowledge the authors’ clarifications. If you have any further questions, feel free to initiate a discussion.

Best,
AC.

---

### Meta-Review · Area_Chair_BFEs · 2026-01-08

**Summary:**

This paper presents an intuitive decomposition-based framework for temporal dense action detection and reports consistent performance gains on benchmark datasets. However, the proposed approach is closely related to prior two-stream and multi-branch methods, and the paper does not sufficiently clarify its novelty or isolate the effect of decomposition from increased model capacity. In addition, concerns remain regarding the robustness of the strategy, fairness and correctness of experimental comparisons, and reproducibility of key components. Overall, the evidence is insufficient to support acceptance. Final recommendation: Reject.

**Reviewer Concerns:**

Resolved

- The rebuttal shows that, targeted ablations comparing semantic sub-labels to “foreground-only” supervision demonstrate that semantic decomposition drives most performance gains.

- Added quantitative results and qualitative visualizations demonstrate the effectiveness and behavior of the two sub-networks.

- Authors provided a consistent argument that the decomposed model need not have more parameters than unified baselines.

- Revisions distinguish semantic sub-label decomposition from classic two-stream or modality-based methods, with fixes to typos and notation confusion.

Unresolved:

- Benchmarking and Fairness Concerns
Skepticism remains about baseline parity, hyperparameter fairness, and independent confirmation of corrected results.

- Novelty Perception Risk
Several reviewers may still view the contribution as incremental, especially regarding claims of being “first” to address overlaps, unless supported by a stronger literature audit.

- Reviewers requested clearer contextual discussion versus recent temporal action localization methods, even if direct comparison is not feasible.

**Reviewer Scores:**

Reviewer Nbui (initial: 4)

All explicit issues they raised were directly addressed (foreground-only ablation, sub-network quantitative + qualitative evidence, corrected tables). This reviewer already sounded “would not mind if accepted,” so stronger evidence likely pushes them slightly above threshold. The reviewer might raise the score slightly.

Reviewer DHHi (initial: 2)

They explicitly stated they retain the rating after seeing the rebuttal. Even with more discussion, their concerns are about fundamental rigor/reproducibility/fairness and they showed no indication of being swayed.  The reviewer might maintain the score slightly.

Reviewer QKDi (initial: 4)

Authors addressed nearly every concrete request: novelty clarification vs named works, label-noise concern, standalone sub-network results, parameter-control argument, and presentation fixes. If the new appendix evidence is strong and clearly written, this reviewer is a good candidate to move slightly above threshold.  The reviewer might raise the score slightly or maintain it.

Reviewer Nh85 (initial: 4)
The reviewer explicitly said they wanted to increase to 5 but couldn’t because the available choices were “4 or 6” around borderline, so they maintained 4. With full participation (and if 5 were available), they would likely have moved up.

---

### Decision · Program_Chairs · 2026-01-26

Reject